# Heterologous Biosynthesis of Health-Promoting Baicalein in *Lycopersicon esculentum*

**DOI:** 10.3390/molecules27103086

**Published:** 2022-05-11

**Authors:** Jingjing Liao, Lei Xie, Tingyao Liu, Changming Mo, Shengrong Cui, Xunli Jia, Xiyang Huang, Zuliang Luo, Xiaojun Ma

**Affiliations:** 1Institute of Medicinal Plant Development, Chinese Academy of Medical Sciences & Peking Union Medical College, Beijing 100193, China; liaojingjing1993@163.com (J.L.); leixie1996@163.com (L.X.); c1061729635@163.com (S.C.); 2College of Horticulture, Shenyang Agricultural University, Shenyang 110866, China; lty1044064148@163.com; 3Guangxi Crop Genetic Improvement and Biotechnology Lab, Guangxi Academy of Agricultural Science, Nanning 530007, China; mochming@126.com; 4Yunnan Key Laboratory of Southern Medicine Utilization, Institute of Medicinal Plant Development Yunnan Branch, Chinese Academy of Medical Sciences, Jinghong 666100, China; xunlijia1998@163.com; 5Guangxi Key Laboratory of Plant Functional Phytochemicals and Sustainable Utilization, Guangxi Institute of Botany, Guangxi Zhuang Autonomous Region and Chinese Academy of Sciences, Guilin 710089, China; xiyanghuanggxib@126.com

**Keywords:** baicalein, multigene vector, *Lycopersicon esculentum*, plant chassis, biofortification, heterologous biosynthesis

## Abstract

Baicalein is a valuable flavonoid isolated from the medicinal plant *Scutellaria baicalensis* Georgi, which exhibits intensive biological activities, such as anticancer and antiviral activities. However, its production is limited in the root with low yield. In this study, In-Fusion and 2A peptide linker were developed to assemble *SbCLL-7*, *SbCHI*, *SbCHS-2*, *SbFNSII-2* and *SbCYP82D1.1* genes driven by the AtPD7, CaMV 35S and AtUBQ10 promoters with HSP, E9 and NOS terminators, and were used to engineer baicalein biosynthesis in transgenic tomato plants. The genetically modified tomato plants with this construct synthesized baicalein, ranging from 150 ng/g to 558 ng/g FW (fresh weight). Baicalein-fortified tomatoes have the potential to be health-promoting fresh vegetables and provide an alternative source of baicalein production, with great prospects for market application.

## 1. Introduction

Baicalein is a kind of flavonoid that is commonly extracted from the roots of *Scutellaria baicalensis* Georgi. It exhibits many pharmacological activities on human health, such as antioxidant activity [1], anti-inflammatory activity [2] and liver protection [3]. More importantly, according to the study, baicalein can commonly induce cancer cell apoptosis without impacting the growth of normal cells, making it a perspective antitumor drug [4,5]. In addition, baicalein was identified as a promising antiviral drug for SARS-CoV-2, which causes COVID-19 infection [6]. Therefore, there is a great market demand for baicalein in pharmaceutical industries. However, the production of baicalein is considered to rely too heavily on the availability of chemical extraction, and baicalein commonly accumulates specifically in roots at a relatively lower level. Currently, the supply of baicalein is insufficient, thus an alternative synthetic production method in heterologous hosts is being intensively explored to meet market demands. [7]. In this case, it is essential to develop an efficient and appropriate technique for baicalein production.

Over the past three decades, major progress in molecular biology and synthetic biology has been made, which has allowed for the reconstruction of metabolic pathways in candidate organisms, and thus has accelerated the elucidation of many natural-product biosynthesis pathways. Comprehensive studies are crucial to our broad understanding of the baicalein biosynthesis pathway, which is produced from cinnamic acid, the common precursor for flavones [8,9]. Cinnamoyl-CoA is formed by the catalysis of cinnamoyl-coenzyme A (CoA) ligase (SbCLL-7). Chalcone synthase (SbCHS-2) is then recruited to form pinocembrin chalcone, and pinocembrin is synthesized by chalcone isomerase (SbCHI). Subsequently, flavone synthase II (SbFNSII-2) is used to convert pinocembrin into chrysin. After this, flavonoid 6-hydroxylases (SbCYP82D1.1) are responsible for baicalein production. The biosynthesis pathway of baicalein is shown in Figure 1. This finding provides a theoretical basis for developing a novel and convenient synthetic approach to engineer baicalein biosynthesis in heterologous hosts. Hence, researchers have shown an increasing interest in the production of baicalein via a microbial chassis. The production of baicalein in *E. coli* with lower yield was first reported by Li et al. in 2018 [7]. Subsequently, researchers have reported the synthesis of baicalein in *S. cerevisiae*, but the level of yield was still low (4.69 mg/L) [10]. Recently, researchers demonstrated that baicalein is synthesized directly from glucose. The researchers showed that the yield increased by approximately 6.6-fold and the titer rose from 21.6 mg/L to 143.5 mg/L [11]. The application of a microbial chassis raises the possibility of baicalein production. Despite microbial chassis being excellent sources of baicalein, this method faces problems of high energy consumption and high cost. As a result, plant extraction remains a great sustainable and practical way for the large-scale production of baicalein. Therefore, engineering the entire biosynthetic pathway in heterologous plants is a promising approach to baicalein production that has several advantages, such as low energy requirements and low waste production. In addition, in experiments on the transient expression of baicalein, five baicalein biosynthesis genes were co-transformed. Furthermore, baicalein was shown to accumulate in *Nicotiana benthamiana* plants [8], which provides considerable theoretical evidence for baicalein production in heterologous plants.

Many published studies have described the importance of the plant chassis in natural-product production, providing a green and efficient synthetic method for producing natural products [12]. Many studies have suggested that the plant chassis provides a more suitable synthetic environment for the heterologous synthesis of plant-based natural products [13]. Over the past two decades, there have been all kinds of secondary metabolites synthesized in heterologous plants for the biofortification of micronutrients, phytonutrients and bioactive components, such as β-carotene, which is synthesized in rice endosperm [14], maize [15], canola [16], potato [17], soybean [18] and banana [19]; the production of anthocyanin in rice endosperm [20], tomato [21] and maize [22]; astaxanthin-enriched rice [23] and tomato [24]; ginsenoside aglycone, which is produced in rice [25]; and the heterologous biosynthesis of artemisinin in *Nicotiana benthamiana* [26] and *Physcomitrella patens* [27]. Metabolic engineering in plant chassis to increase the nutritional traits of crops, vegetables and fruits is a growing field. In a previous study, we transformed mogroside biosynthesis in cucumber, tomato, tobacco and *Arabidopsis thaliana* via multigene vector transformation, which provided a strong theoretical framework for baicalein production in a plant chassis. This approach is not only a potentially fruitful avenue for producing valuable bioactive compounds, but also represents a further step towards developing health-promoting biofortified plants. In short, this study can contribute to a better understanding of health-promoting functional and nutritious vegetable breeding and may provide new insights into the industrial-scale production of numerous metabolites.

To achieve the heterologous biosynthesis of baicalein, the most challenging problem is the simultaneous expression of the five baicalein biosynthesis genes in a single heterologous plant chassis. In terms of multigene vector construction, we combined In-Fusion technology and self-cleaving 2A peptides to assemble five baicalein biosynthesis genes into pCAMBIA1300, which included *SbCLL-7*, *SbCHI*, *SbCHS-2*, *SbFNSII-2* and *SbCYP82D1.1* genes. The genetic transformation system of Micro-Tom tomato was successfully established by *Agrobacterium tumefaciens* with the introduction of baicalein biosynthesis genes, which had a higher transcription level in the transgenic plants. For the first time, the present research explores whether baicalein-fortified transgenic tomato plants have the potential to produce baicalein by chemical extraction. This study sheds new light on the genetic improvement of health-promoting biofortified vegetables and may have practical implications for baicalein production.

## 2. Results

### 2.1. Design of Multigene Expression Vector

In this study, a multigene expression vector was assembled with *SbCYP82D1.1*, *SbCLL-7*, *SbCHI*, *SbCHS-2*, *SbFNSII-2* and *Hyg* genes driven by the AtPD7, CaMV 35S and AtUBQ10 promoter (Figure 2A). A schematic representation of the multigene assembly strategy is shown in Appendix A. The strategy consisted of four rounds of multigene assembly. Firstly, each candidate gene was successfully isolated from *Scutellaria baicalensis* cDNA, and *SbCYP82D1.1* was inserted into PBI121 with the *AtPD7* promoter and HSP terminator at *Hin*dIII and *Eco*RI. Furthermore, to produce the double gene expression cassette, *SbCLL-7* and *SbCHI* were linked by self-cleaving 2A peptides and inserted into pCAMBIA1300 with the *CaMV 35S* promoter and E9 terminator at the *Eco*RI and *Hin*dIII restriction sites. Similarly, *SbCHS-2* and *SbFNSII-2* were fused with P2A peptides, which were inserted into pCAMBIA1300 with the AtUBQ10 promoter and NOS terminator. Then, the promoter, target gene and terminator were cloned and ligated into pCAMBIA1300 to produce AtPD7-SbCYP82D1.1-Thsp-35S-SbCLL-7-2A-SbCHI-Te9. Then, the regions of AtPD7-SbCYP82D1.1-Thsp-35S-SbCLL-7-2A-SbCHI-Te9 and AtUBQ10-SbCHS-2-2A-SbFNSII-2-Tnos were ligated into the final multigene expression vector. The recombination plasmid FCC-CF was identified by DNA sequencing. The inserted sequence of recombination plasmid FCC-CF is listed in Appendix A.

### 2.2. Genetic Transformation of Tomato Plants and Molecular Analysis

The FCC-CF vector was introduced into *Agrobacterium tumefaciens* to synthesize baicalein in engineered tomato plants using plant genetic transformation (Figure 2A). Three independent transgenic lines F8, F12 and F21 were screened using hygromycin resistance and were cultivated in the greenhouse (Figure 2B). Genomic PCR was used to detect the integration of five baicalein biosynthesis genes into the genome of transgenic tomato plants. The 1554-bp fragment of *SbCYP82D1.1*, the 1633-bp fragment of *SbCLL-7*, the 648-bp fragment of *SbCHI*, the 1170-bp fragment of *SbCHS-2*, the 1518-bp fragment of *SbFNSII-2* and the 392-bp fragment of the *Hyg* gene were simultaneously detected in the transgenic Micro-Tom tomato plants F8, F12 and F21, which was in accordance with the positive plasmid (FCC-CF) (Figure 2C). The five corresponding fragments above were also detected in transgenic tomato lines, including F8, F12 and F21, but were not detected in wild-type Micro-Tom plants. The growing conditions of transgenic plants were consistent with non-transformed tomato plants. Then, the relative transcription levels of the five baicalein biosynthesis genes were detected using qRT-PCR. The transgenic Micro-Tom tomato lines showed significantly higher expression of *SbCLL-7*, *SbCHI*, *SbCHS-2*, *SbFNSII-2* and *SbCYP82D1.1* genes compared to the WT Micro-Tom tomato plants (Figure 3). The PCR and qRT-PCR analysis indicated that the baicalein biosynthesis pathway, consisting of *SbCYP82D1.1*, *SbCLL-7*, *SbCHI*, *SbCHS-2* and *SbFNSII-2*, were expressed in transgenic tomato lines.

### 2.3. HPLC-MS/MS Analysis for Baicalein in Transgenic Tomato Lines

To acquire optimal MS conditions for the identification and quantification of baicalein in the samples, a standard solution (50 ng/mL) of baicalein was directly introduced into the mass spectrometer through the manual tuning mode using a syringe pump (Harvard Apparatus, Australia). In the full-scan Q1 mass spectrum, the parent positive ion peak [M+H]+ appeared at *m/z* 271.0. Thus, *m/z* 271.0 was targeted for fragmentation under product ion mode, the MS/MS spectra generated the fragment ions *m/z* 253.1, 169.0, 150.9 and 122.9 that were characteristic of baicalein (Figure 4A). Then, the most stable and abundant ions *m/z* 169.0 and 122.9 were selected for confirmation and quantification. The production of transgenic tomato fruits was analyzed under the multiple reaction monitoring (MRM) mode by HPLC-MS/MS. The total ion chromatogram (TIC) revealed that a new compound was found in the transgenic tomato lines F12 and F21 at the retention time of 3.65 min, which was consistent with the retention time of the baicalein standard (Figure 4B). According to the retention time and MS/MS transitions, we confirmed that the compound was baicalein, which was synthesized in the transgenic Micro-Tom tomato plants. Then, the average baicalein contents were measured in the transgenic Micro-Tom tomato lines, which were 150 ng/g and 558 ng/g FW in F12 and F21, respectively (Figure 5). As expected, baicalein was not detected in the wild-type Micro-Tom tomato plant fruits. In addition, baicalein was not found in the F8 transgenic line. Baicalein production in the transgenic tomato plants demonstrated that the multigene vector FCC-CF harboring five baicalein biosynthesis genes was successfully simultaneously expressed in the transgenic Micro-Tom tomato plants F12 and F21. This study has provided the possibility to generate novel health-promoting tomato germplasms with baicalein fortification.

## 3. Discussion

As a fresh vegetable for consumption, tomato has been widely grown on a global scale with high yield. Previous researchers have suggested that all kinds of bioactive compounds are present in tomato plants, including flavonoids, terpenoids, polyphenolics and carotenoids, which may be used as promising intermediates and substrates for the biosynthesis of various metabolites [28,29]. Accordingly, Micro-Tom tomato plants have a shorter growth cycle and a stable genetic transformation system, which provides an excellent plant chassis for heterologous metabolite production [30]. As a model vegetable crop, the tomato has an enormous amount of innovative scientific biotechnological application. In recent years, researchers have shown an increased interest in the production of high-value and health-promoting secondary metabolites in tomato plants, which have led to increasingly rapid advances, such as betanin [31], L-DOPA [32], anthocyanin [21], astaxanthin [24] and rosmarinic acid [33], etc. In this study, Micro-Tom tomato was used to produce baicalein with the transformation of five baicalein biosynthesis genes. Although the content was relatively lower, this study is of great significance as it marks the first attempt to produce baicalein in transgenic tomato plants. This research can be used to address the problem of lower accumulation of baicalein in the original plant and has important implications for baicalein production without *Scutellaria baicalensis*. This study aims to explore the potential for baicalein production in heterologous plant hosts. Further study with a greater focus on health-promoting biofortified tomato development is needed, as it is a significant factor contributing to potential anticancer and anti-virial drug development.

Technically, to achieve the production of baicalein in heterologous hosts, an available multigene assembly strategy was crucial. Traditionally, retransformation and co-transformation methods were used to transmute multiple genes into heterologous plant chassis for their introduction to complete metabolic pathways. However, these methods are time-consuming, laborious and are not applicable for the transformation of multiple genes simultaneously [34]. Thus, multigene stacking provided a significant advantage, which integrated multiple gene expression cassettes into the host chromosome genome with a single T-DNA insertion [35]. Over the past decade, there has been rapid development in multigene stacking, such as suppression thermo-interlaced PCR (STI PCR) [36], Gibson assembly [37], Cre recombinase/loxP-mediated recombination (TransGene Stacking II, TGS II) [20,23], GATEWAY [38] and In-Fusion [39]. TGS II is an efficient method for multiple gene stacking, which has transformed 4–8 genes into rice endosperm. However, patent issues have limited this promising technology that, ultimately, has not been widely adopted in the production of diverse natural products. Moreover, GATEWAY has a repetitive sequence in the multiple DNA sequence splicing, which has an adverse effect on gene expression. Interestingly, Gibson assembly and In-Fusion have been widely used to assemble multiple overlapping DNA fragments simultaneously into a binary vector. In practice, however, an In-fusion-based Gene Stacking (IGS) strategy was developed for the transgene stacking of a multigene vector harbouring five baicalein biosynthesis genes and was transformed into tomato plants. Gibson assembly was not available for this study. The nucleotide sequence of insertion fragments was too long to be suitable for Gibson assembly. Previous studies have demonstrated that attempts to increase the number of transgenes in a vector allow more unstable vectors, thus leading to gene silencing. In this case, it is important to consider the effectiveness of the multigene vector. Therefore, in this study, to avoid the consequence of repeated sequences, P2A peptide was used to ligate *SbCYP82D1.1*, *SbCLL-7*, *SbCHI*, *SbCHS-2* and *SbFNSII-2*, and three promoters were used to express the five baicalein biosynthesis genes efficiently in transgenic plants. We hypothesized that the cinnamic acid in the target plant could be used to transfer this multigene vector for baicalein production. This study describes extensive prospective applications in the field of biofortified crop improvement and shows the potential for the industrial-scale production of baicalein.

Since molecular detection indicated that the baicalein biosynthetic pathway was introduced into the Micro-Tom tomato plants, it was interesting that *S**bCLL-7*, *SbCHI*, *SbCHS-2*, *SbFNSII-2* and *SbCYP82D1.1* genes did not exist in Micro-Tom tomato plants, but the above five genes were simultaneously detected in the transgenic lines. Furthermore, the expression levels of *SbCHI*, *SbCHS-2* and *SbFNSII-2* genes increased at least 7-fold in the F21 transgenic line compared to the WT Micro-Tom tomato plants and other target genes were up-regulated from 10 to 35,000-fold. Generally, the transcripts of the target genes were not detected in wild-type plants, but were detected in the transgenic tomato lines. Therefore, the expression levels of these target genes were considered to be over-expressed in the transgenic lines. It was interesting to assess the nootkatone production in plants, notably, when the baicalein biosynthesis genes were expressed and baicalein accumulation was assessed using HPLC-MS/MS. As expected, baicalein was produced at relatively lower levels in the F12 and F21 transgenic lines. However, baicalein was not detected in the F8 transgenic line, which had a higher expression level of baicalein biosynthesis genes. The baicalein contents were significantly different among the three transgenic lines, indicating that the insertion position of transgenes may affect baicalein production. Generally, the insertion position plays a critical role in gene expression regulation. The T-DNA region with multiple gene expression cassettes was inserted into the genome randomly in the plant genetic transformation, which has greatly decreased the efficiency of the integration, hereditation and expression of external genes in transgenic plants. The random insertion into the transgenic plant genome was potentially harmful to the copy number of transgenes, protein activities and inactivated enzymes. Additionally, in a plant cell, there are thousands of proteins, and it is uncommon for each protein to perform its function independently. Furthermore, protein interactions may inhibit the production of baicalein in transgenic plants. Therefore, our study lays a crucial foundation for baicalein production in engineered tomato plants. However, the function of proteins in transgenic plants and the production of baicalein at relatively higher levels still need further research.

In addition, previous studies showed that baicalein exhibited significant antiviral activity against COVID-19 [40], dengue virus [41], Chikungunya virus [4], influenza virus [42] and herpes simplex virus type 1 [43], which may improve human health through daily diet. In addition, transgenic tomato plants have stronger disease resistance than WT plants, which may improve the antiviral activity of tomatoes. However, the agronomic traits and medicinal value of transgenic tomato plants containing baicalein should be further assessed. In summary, we demonstrated that the baicalein biosynthesis pathway can be reconstructed in the transgenic tomato through the introduction of five baicalein biosynthesis genes. The baicalein-fortified tomato can be used as a functional fresh-eating vegetable to improve one’s health. Furthermore, it provides valuable insight into the development of raw material for baicalein production.

## 4. Materials and Methods

### 4.1. Plant Material, Chemicals and Strains

*Lycopersicon esculentum* (Micro-Tom tomato) was used for plant transformation. *Escherichia coli* strains DH5α, XL10-Gold and *Agrobacterium tumefaciens* strain GV3101 (WeidiBio, Shanghai, China) were used in this experiment.

HPLC-grade formic acid, methanol and acetonitrile were purchased from Fisher (Emerson, IA, USA). ClonExpress Ultra One Step Cloning Kit, HiScript III 1st Strand cDNA Synthesis Kit (+gDNA wiper) and Taq Pro Universal SYBR qPCR Master Mix were purchased from Vazyme Biotech Co., Ltd. (Nanjing, China). Ultrapure RNA Kit (DNase I) was obtained from CWBIO. Co., Ltd. (Beijing, China). Plant genomic DNA kits, restriction enzymes and DNA Marker II were purchased from TianGen Biotech Co., Ltd. (Beijing, China). KOD One PCR master Mix was obtained from TOYOBO Biotech Co., Ltd. (Shanghai, China). The PBI121 and pCAMBIA1300 plasmids were from laboratory stock. The baicalein standard was obtained from Shanghai Yuanye Bio-Technology Co., Ltd. (Shanghai, China). Other reagents were purchased from Beijing Chemical Corporation (Beijing, China) unless otherwise specified.

### 4.2. Multigene Vector Construction

To construct the multigene expression vector, the full-length sequences of the *SbCLL-7*, *SbCHI*, *SbCHS-2*, *SbFNSII-2* and *SbCYP82D1.1* genes were amplified from the leaf cDNAs of *Scutellaria baicalensis* by PCR amplification. The *AtUBQ10* and *AtSCPL30* promoters were cloned from *Arabidopsis thaliana* using KOD One PCR master Mix. These promoters have previously been used in apples [44] and tobacco [45]. Then, the constitutive promoter, CaMV 35S promoter, and the nopaline synthase (NOS) terminator were amplified from the PBI121 plasmid. The terminator from the pea rbcS-E9 gene (Te9) and heat-shock protein (HSP) 18.2 terminator were chemically synthesized by GENEWIZ (GENEWIZ, Suzhou, China).

In round I of multigene assembly, *SbCYP82D1.1* was fused with the *AtPD7* promoter and HSP terminator to form AtPD7-SbCYP82D1.1-Thsp. In addition, *SbCLL-7* and *SbCHI* were ligated into PBI121 with a 2A peptide linker from porcine teschovirus (P2A) (the amino acid sequence was GSGATNFSLLKQAGDVEENPGP) via a ClonExpress Ultra One Step Cloning Kit. Similarly, *SbCHS-2* and *SbFNSII-2* were fused into PBI121 at *Bam*HI/*Sac*I sites. For round II of multigene assembly, the CaMV 35S promoter, the sequence of SbCLL-7-P2A-SbCHI and the terminator from the pea rbcS-E9 gene were ligated into the pCAMBIA1300 plasmid to generate 35S-SbCLL-7-P2A-SbCHI-Te9. The region of SbCHS-2-2A-SbFNSII-2 was amplified from the recombination vector. Then, the AtUBQ10 promoter, SbCHS-2-2A-SbFNSII-2 and NOS terminator were inserted at the *Eco*RI and *Hin*dIII restriction sites to generate AtUBQ10-SbCHS-2-2A-SbFNSII-2-Tnos. Subsequently, in round III of multigene assembly, the regions of AtPD7-SbCYP82D1.1-Thsp and 35S-SbCLL-7-P2A-SbCHI-Te9 were amplified and inserted into pCAMBIA1300 at the *Eco*RI/*Hin*dIII restriction sites, which were able to harbor three baicalein biosynthesis genes. In round IV of multigene assembly, AtPD7-SbCYP82D1.1-Thsp-35S-SbCLL-7-P2A-SbCHI-Te9 and AtUBQ10-SbCHS-2-2A-SbFNSII-2-Tnos were cloned and ligated into the pCAMBIA1300 plasmid via the ClonExpress Ultra One Step Cloning Kit, yielding a multigene expression cassette named FCC-CF (Appendix A). The multigene vector FCC-CF was transferred into *Agrobacterium tumefaciens* GV3101 via the freeze-thaw method. Then, FCC-CF was transformed into Micro-Tom tomatoes to generate baicalein-fortified tomato transgenic lines. All of the primers for multigene assembly are listed in Appendix A.

### 4.3. Agrobacterium-Transformation of Micro-Tom Tomato Plants

Micro-Tom tomato seeds were washed using sterile water and were sterilized by 75% ethanol and 10% NaClO for 30 s and 10 min, respectively. Then, sterile water was used to wash the seeds five times, and the seeds were dried by sterilized filter paper. All seeds were sown on 1/2 MS media. In this study, cotyledons and hypocotyls were explants, which were cut and incubated for 30 min with *Agrobacterium* harboring the multigene expression vector FCC-CF, and were then co-cultivated in MS media containing indole-acetic acid (IAA) (0.2 mg/L) and zeatin (ZT) (2.0 mg/L). Then, the explants were cultivated in selective medium, which was MS media with 0.2 mg/L IAA, 2.0 mg/L ZT, 500 mg/L carbenicillin (carb) + 10 mg/L hygromycin (hygr). After that, the shoots were transferred into elongation media (MS media + 0.05 mg/L IAA + 1.0 mg/L ZT + 500 mg/L carb + 10 mg/L hygr). Regenerated plants were subsequently transferred into the rooting culture media consisting of 1/2 MS media + 0.1 mg/L IAA + 500 mg/L carb + 2 mg/L hygr. The transgenic tomato plants were grown in the greenhouse under a 16 h light/8 h dark photoperiod.

### 4.4. PCR Detection of Transgenic Tomato Plants

To detect the integration of transgenes in the genome of transgenic tomato plants, the leaves of resistant plant DNA were obtained using a Plant Genomic DNA Kit. This was used as the template. KOD One PCR Master Mix was used to verify the *SbCYP82D1.1*, *SbCLL-7*, *SbCHS-2*, *SbCHI*, *SbFNSII-2* genes and hygromycin resistance gene (*Hyg*) in the transgenic plants using the specific primers. The recombination vector FCC-CF was a positive control. As the negative control, wild-type (WT) plant DNA was extracted. All of the primers used in PCR detection are shown in Appendix A.

### 4.5. Quantitative Real-Time PCR (qRT-PCR) Analysis

To detect the relative expression levels of *SbCYP82D1.1*, SbCLL-7, *SbCHI*, *SbCHS-2* and *SbFNSII-2* genes in transgenic tomato plants, qRT-PCR was performed in this study. To normalize the gene expression level, the ACTIN gene from *Lycopersicon esculentum* was selected as a reference gene. The total RNA from the transgenic plants and wild-type plants were obtained via the Ultrapure RNA Kit (DNase I). In addition, cDNA was synthesized by HiScript III 1st Strand cDNA Synthesis Kit (+gDNA wiper). qRT-PCR analysis was performed using a Taq Pro Universal SYBR qPCR Master Mix with an ABI CFX96TM Real-Time System (Waltham, Massachusetts, USA). The real-time PCR conditions were 95 °C for 30 s, followed by 40 cycles of 95 °C for 3 s and 55 °C for 10 s. 2^−ΔΔCt^ method was used to calculate the relative expression levels. All data obtained from the three independent experiments were shown as the mean ± SD. All of the primers used for the qRT-PCR analysis are shown in Appendix A. 

### 4.6. Analysis of Baicalein by HPLC-MS/MS

Baicalein accumulation was analyzed using HPLC-MS/MS. First, 1.5 g of the fruits of transgenic tomato plants and wild-type tomato plants were powdered in liquid nitrogen. All samples were soaked in 70% methanol and were ultrasonically extracted for 1 h. The supernatant was then collected by centrifugation at 5000× *g* for 20 min and was filtered with a 0.22 µM Millipore filter.

AB Sciex 4500 QTRAP LC/MS/MS (Toronto, ON, Canada) and an Agilent Technologies 1260 Series LC system (Agilent, Santa Clara, CA, USA) were used for HPLC-MS/MS analysis. Agilent Poroshell 120 SB C18 column (100 mm × 2.1 mm, 2.7 µm) was equipped for this instrument. The mobile phase A was water (0.1% formic acid) and the mobile phase B was acetonitrile. The gradient elution was used to analyze the baicalein. The HPLC conditions were as follows: 0 min for 20% B; 0.5 min for 30% B; 2–4 min for 98% B; and 5.5–8 min for 20% B. The flow rate was 0.3 mL/min. The electrospray ionization (ESI) source was performed in positive ion mode. The detection of baicalein was performed in an optimal multiple reaction monitoring (MRM) mode, with transitions of [M+H]+ *m/z* 271.0→122.9 for quantification and 271.0→169.0 for confirmation. For each MRM transition, the dwell time was 200 ms. The declustering potential (DP) and collision energy (CE) were set at 30 V and 30 eV, respectively. The baicalein standard was dissolved in methanol. Each experiment was performed three times.

### 4.7. Data Analysis

All samples were randomly collected in this experiment. The data was analyzed using SPSS 16.0. Origin 2019b (OriginLab Co., Northampton, MA, USA) was used to graph the data.

## Figures and Tables

**Figure 1 molecules-27-03086-f001:**
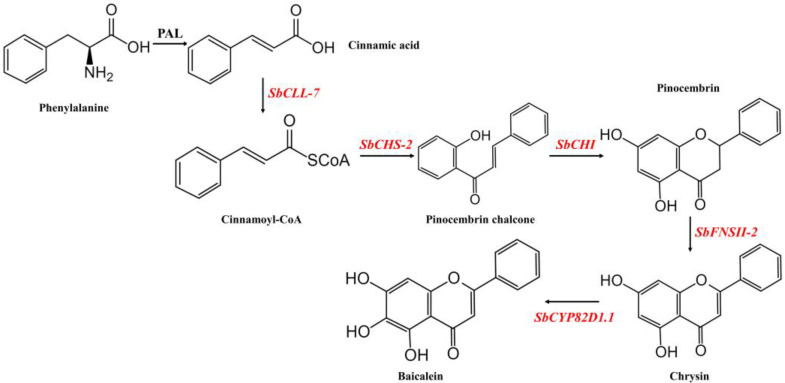
Map of the engineered baicalein biosynthesis pathway in *Lycopersicon esculentum*. Baicalein biosynthesis genes that were transformed into tomatoes in this study are marked in red. The five genes involved in the baicalein biosynthetic pathway are *SbCLL-7*, cinnamoyl-coenzyme A (CoA) ligase; *SbCHS-2*, chalcone synthase; *SbCHI*, chalcone isomerase; *SbFNSII-2*, flavone synthase II and *SbCYP82D1.1*, flavonoid 6-hydroxylases, respectively.

**Figure 2 molecules-27-03086-f002:**
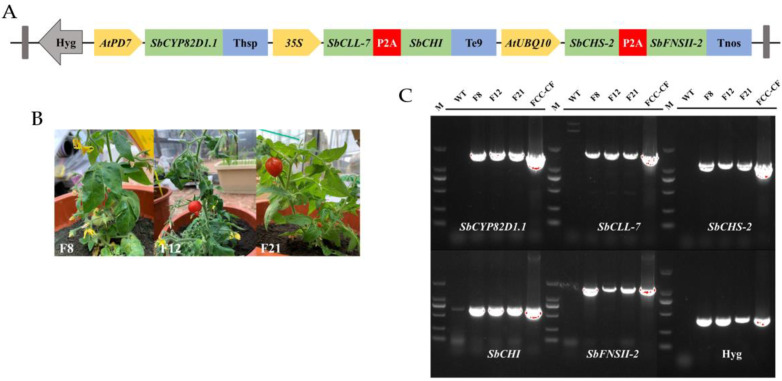
*Agrobacterium* transformation of Micro-Tom tomato plants with baicalein biosynthesis genes. (**A**) T-DNA region of the multigene vector FCC-CF that was transformed into the Micro-Tom tomato plants. (**B**) Transgenic Micro-Tom lines (F8, F12 and F21). (**C**) PCR detection of five baicalein biosynthesis genes and Hyg genes. The wild-type (WT) was not transformed with *Agrobacterium*. F8, F12 and F21 were transgenic Micro-Tom lines. FCC-CF was the recombination plasmid. M represented the 2000 bp maker.

**Figure 3 molecules-27-03086-f003:**
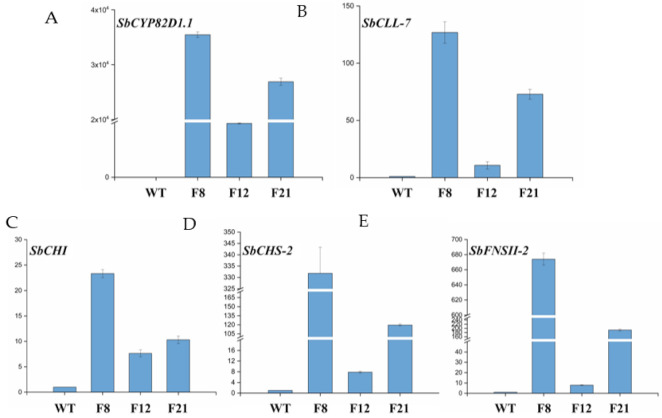
The relative gene-expression-level analysis of candidate genes in the transgenic Micro-Tom tomato plants and WT plants. (**A**) The relative expression-level analysis of *SbCYP82D1.1*. (**B**) The relative expression-level analysis of *SbCLL-7*. (**C**) The relative expression-level analysis of *SbCHI*. (**D**) The relative expression-level analysis of *SbCHS-2*. (**E**) The relative expression-level analysis of *SbFNSII-2*. WT: wild-type Micro-Tom tomato plants. F8, F12 and F21: transgenic tomato plants. *Leactin* gene from *Lycopersicon esculentum* was used as the reference gene. The data are represented as mean ± SD for the three independent experiments.

**Figure 4 molecules-27-03086-f004:**
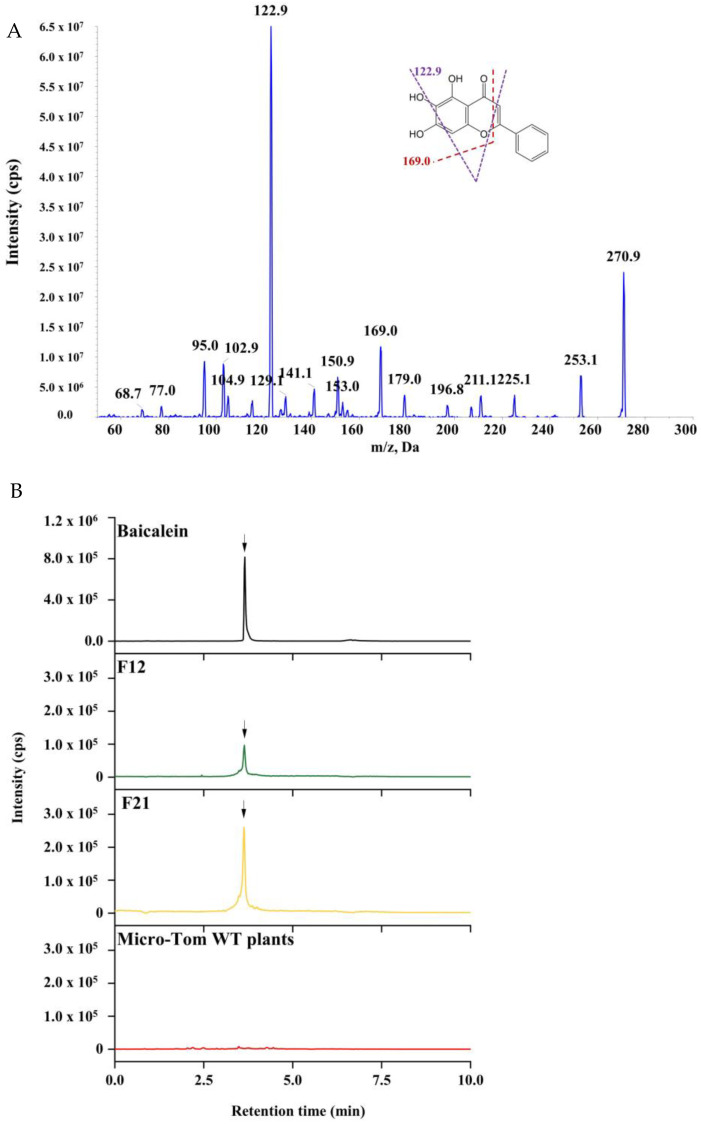
HPLC-MS/MS analysis of baicalein in transgenic Micro-Tom tomato lines. (**A**) Full-scan product ion spectra and the proposed fragmentation schemes of baicalein. (**B**) The total ion chromatogram (TIC) of the samples and the control, respectively.

**Figure 5 molecules-27-03086-f005:**
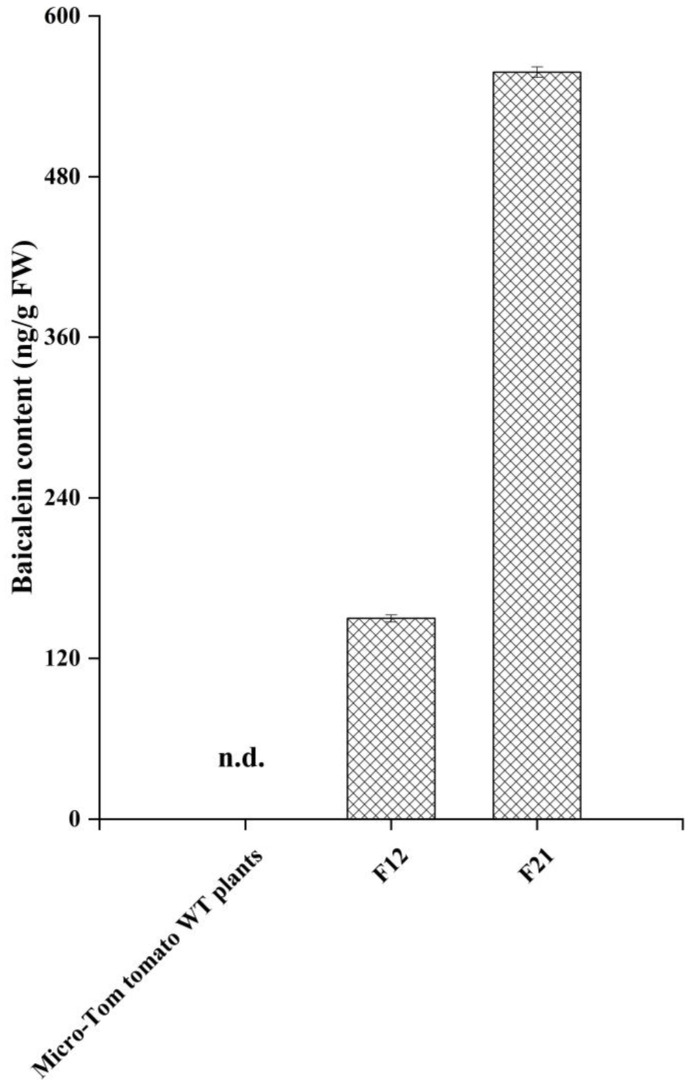
Average baicalein content (ng/g fresh weight) in the fruits of transgenic Micro-Tom tomato lines. n.d.: not detected. The data are represented as mean ± SD for the three independent experiments.

## Data Availability

The authors confirm that all data in this experiment are available in the main text and Appendix A.

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
