# Peer review of "Heterologous Biosynthesis of Health-Promoting Baicalein in *Lycopersicon esculentum"

_molecules, 2022, doi:10.3390/molecules27103086_

Round 1
Reviewer 1 Report
Abstract:
The Abstract and also the introduction lack necessary information regarding the importance of the current research and the authors do not follow a clear aim and hypothesis.
The Abstract should capture the main points of the paper. The Abstract should be focusing on the following: What was investigated (state clearly the specific objective(s) of the study); Why was it necessary; Briefly state how was it done; What are the main results; What is the value of the study. Superfluous sentences and irrelevant statements should be eliminated.
Results:
The authors relied on the comparison with the baicalein standard to verify that the experiment showed heterologous expression of the set of genes in the host plant. Great, very good strategy! However, if the HPLC-MS/MS technique was used for the extracts analysis, I really missed the presentation of mass spectra to confirm the molecular structure of the metabolite of interest.
Although there is great specificity of the genetic machinery involved in baicalein biosynthesis, I REALLY believe that the presentation of the fragmentation mechanisms (MS/MS) will exclude any possibility that the hydroxyl of the C6 position (baicalein structure) is not located in the C8 position. I suggest to include, in the supplementary material, the mass spectra, as well as the fragmentation proposal for the molecule in question.
Author Response
Dear reviewer,
Thank you for your letter and for your comments concerning our manuscript entitled " Heterologous biosynthesis of health-promoting baicalein in Lycopersicon esculentum", which submitted to molecules. Those comments are all valuable and very helpful for revising and improving our paper, as well as the important guiding significance to our researches. We have studied comments carefully and have made correction which we hope meet with approval. The main corrections in the paper and the responds to the reviewer’s comments are as flowing:
Response to comment:
- We have modified the abstract and introduction in the mauscript according to your suggestion.
- We have added the mass spectra to confirm the molecular structure of the baicalein in the Figure 4, which has been interpreted in the result 2.3.
We would like to express our great appreciation to your comments on our paper. Looking forward to hearing from you.
Thank you and best regards.
Yours sincerely,
Corresponding author: Xiaojun Ma
Reviewer 2 Report
The manuscript “Heterologous biosynthesis of health-promoting baicalein in Lycopersicon esculentum” by Liao et al. reports baicalein biosynthesis genes were overexpressed in the transgenic tomato lines F12 and F21 and the average baicalein content was 150 ng/g and 558 ng/g FW (fresh weight), respectively. Although authors reported preliminary the average baicalein content, some important results are still lack from this manuscript. Therefore, I would suggest authors may take at least a major revision before publication. Here are the comments and suggestions:
- In the results, three transgenic tomato lines (including F8, F12, F21) were mentioned, but only two of them were discussed?
- The overexpression in Figs. 2(C) and 2(D) are disagreed.
- The results of F8 line should be added in Figs. 3 and 4, and then discussed.
- The scale of Y-axis in Fig. 3 should be the same for easier comparison.
- Some intermedia should be also collected and analysis, please see Metabolic Engineering 52, 124-133, 2019 and ACS Synth. Biol. 10(5), 1087–1094, 2021.
Author Response
Dear reviewer,
Thank you for your letter and for your comments concerning our manuscript entitled " Heterologous biosynthesis of health-promoting baicalein in Lycopersicon esculentum", which submitted to molecules. Those comments are all valuable and very helpful for revising and improving our paper, as well as the important guiding significance to our researches. We have studied comments carefully and have made correction which we hope meet with approval. The main corrections in the paper and the responds to the reviewer’s comments are as flowing:
Response to comment:
- In the results, three transgenic tomato lines (including F8, F12, F21) were mentioned, but baicalein were found in the transgenic lines F12 and F21, but F8 was not detected the baicalein. So F8 was not discussed.
- SbCLL-7, SbCHI, SbCHS-2, SbFNSII-2 and 1 genes were not existed in Lycopersicon esculentum, and these five genes were detected in the transgenic lines according to the Fig2C, which was the PCR detection. And in the Fig 2D, the expression levels of SbCHI, SbCHS-2 and SbFNSII-2 genes increased at least 7-fold in F21 transgenic line compared to the WT Micro-Tom tomato plants, and other target genes were up-regulated for 10 to 35000-fold. In the case, the expression levels of these target genes were considered to be over-expressed in the transgenic lines.
- Baicalein was not found in the transgenic lines F8, so F8 was not be added in the Fig.3 and Fig.4. But according to the suggestion, we have been discussed the F8 line in the discussion.
- The scale of Y-axis in Fig. 3 was modified and the same Y-axis have been added in the all samples for easier comparison, except for standard, because the peak height was very high in the baicalein standard.
- We have tried to detected some intermediate products, however, unluckily, we haven’t found any intermediate products in the transgenic probably due to the lower content.
We would like to express our great appreciation to your comments on our paper. Looking forward to hearing from you.
Thank you and best regards.
Yours sincerely,
Corresponding author: Xiaojun Ma
Reviewer 3 Report
This interesting manuscript describes the enhanced production of baicalein using the transgenic tomato lines. This study is recommended for publication after addressing the below minor issues.
- Please explain in more detail how to identify baicalein among the compounds with the same molecular weight as baicalein using HPCL-MS/MS analysis.
- Page 2, line 74: “And” should be “and.”
- Page 2, lines 77-78: The following sentence is overlapped and thus should be removed. “There is evidence that the possibility of baicalein production in heterologous plants.”
- Page 4, Figure 2: Please use larger letters.
- Page 7, line 219: Please remove “such as.”
Author Response
Dear reviewer,
Thank you for your letter and for your comments concerning our manuscript entitled " Heterologous biosynthesis of health-promoting baicalein in Lycopersicon esculentum", which submitted to molecules. Those comments are all valuable and very helpful for revising and improving our paper, as well as the important guiding significance to our researches. We have studied comments carefully and have made correction which we hope meet with approval. The main corrections in the paper and the responds to the reviewer’s comments are as flowing:
- Firstly, we relied on the comparison with the baicalein standard to verify that baicalein in the transgenic lines. And then characteristic ion pair to identify the same molecular weight as baicalein using HPCL-MS/MS analysis. Additionally, the specificity of enzyme excludes the possibility of isomers in this study. Moreover, we have added the mass spectra to confirm the molecular structure of the baicalein in the Figure 4, which has been interpreted in the result 2.3.
- Page 2, line 74: The dot has been added in front of “And”.
- Page 2, lines 77-78: “There is evidence that the possibility of baicalein production in heterologous plants.” has been removed.
- Page 4, Figure 2 have been divided into the Figure 3 and Figure 4. And larger letters were added in the figure.
- Page 7, line 219: “such as.” has been removed.
We would like to express our great appreciation to you and reviewers for comments on our paper. Looking forward to hearing from you.
We would like to express our great appreciation to your comments on our paper. Looking forward to hearing from you.
Thank you and best regards.
Yours sincerely,
Corresponding author: Xiaojun Ma
Round 2
Reviewer 2 Report
It seems more acceptable now.